

# Isolation of wheat bran-colonizing and metabolizing species from the human fecal microbiota

Kim De Paepe[1], Joran Verspreet[2,8], Mohammad Naser Rezaei[2], Silvia Hidalgo Martinez[3], Filip Meysman[3,4], Davy Van de Walle[5], Koen Dewettinck[5], Jeroen Raes[6,7], Christophe Courtin[2] and Tom Van de Wiele[1]

[1] Faculty of Bioscience Engineering, Department of Biotechnology, Center for Microbial Ecology and Technology (CMET), Universiteit Gent, Gent, Belgium
[2] Faculty of Bioscience Engineering, Leuven Food Science and Nutrition Research Centre (LFoRCe), Laboratory of Food Chemistry and Biochemistry, KU Leuven, Heverlee, Belgium
[3] Faculty of Sciences, Department of Biology, Ecosystem Management Research Group (ECOBE), Universiteit Antwerpen, Antwerpen, Belgium
[4] Department of Biotechnology, Delft University of Technology, Delft, The Netherlands
[5] Faculty of Bioscience Engineering, Department of Food Technology, Safety and Health, Laboratory of Food Technology and Engineering (FTE), Universiteit Gent, Gent, Belgium
[6] Department of Microbiology and Immunology, Rega Institute, KU Leuven, Leuven, Belgium
[7] Center for Microbiology, VIB, Leuven, Belgium
[8] Current affiliation: Flemish Institute for Technological Research (VITO), Mol, Belgium

Corresponding author
Tom Van de Wiele,
Tom.VandeWiele@UGent.be

## ABSTRACT

Undigestible, insoluble food particles, such as wheat bran, are important dietary constituents that serve as a fermentation substrate for the human gut microbiota. The first step in wheat bran fermentation involves the poorly studied solubilization of fibers from the complex insoluble wheat bran structure. Attachment of bacteria has been suggested to promote the efficient hydrolysis of insoluble substrates, but the mechanisms and drivers of this microbial attachment and colonization, as well as subsequent fermentation remain to be elucidated. We have previously shown that an individually dependent subset of gut bacteria is able to colonize the wheat bran residue. Here, we isolated these bran-attached microorganisms, which can then be used to gain mechanistic insights in future pure culture experiments. Four healthy fecal donors were screened to account for inter-individual differences in gut microbiota composition. A combination of a direct plating and enrichment method resulted in the isolation of a phylogenetically diverse set of species, belonging to the *Bacteroidetes*, *Firmicutes*, *Proteobacteria* and *Actinobacteria* phyla. A comparison with 16S rRNA gene sequences that were found enriched on wheat bran particles in previous studies, however, showed that the isolates do not yet cover the entire diversity of wheat-bran colonizing species, comprising among others a broad range of *Prevotella*, *Bacteroides* and *Clostridium* cluster XIVa species. We, therefore, suggest several modifications to the experiment set-up to further expand the array of isolated species.

## INTRODUCTION

Gut microbiome research has recently started to focus on the microbial composition and functionality of distinct gut environments, such as the mucus layer (*Belzer & De Vos, 2012*; *Bhat et al., 1980*; *Bollinger et al., 2007a*; *Bollinger et al., 2007b*; *Macfarlane & Dillon, 2007*; *Nava, Friedrichsen & Stappenbeck, 2011*; *Probert & Gibson, 2002*; *Swidsinski et al., 2008*; *Van den Abbeele et al., 2013*; *Van den Abbeele et al., 2012*). The importance of undigested, insoluble food particles as microbial colonization sites and their impact on functionality, however, has been poorly studied (*Macfarlane, Hopkins & Macfarlane, 2011*). Results obtained by *Walker et al. (2008)* and *Macfarlane, McBain & Macfarlane (1997)*, *Macfarlane & Macfarlane (2006)* are inconclusive with respect to the existence of a distinct microbial community associated with the particulate matter in fecal samples. *Leitch et al. (2007)* previously pinpointed a specific colonization pattern of insoluble substrates in an anaerobic fermentor system, with wheat bran colonization being dominated by members of *Clostridium* cluster XIVa and *Bacteroides* species (*Leitch et al., 2007*). In an attempt to shed more light on the specific colonization of plant polysaccharides present in the human diet, we have previously performed a series of experiments using wheat bran as an insoluble model substrate. Static batch incubations and experiments in the Simulator of the Human Intestinal Microbial Ecosystem (SHIME) confirmed the colonization of wheat bran by a specific subset of gut bacteria, comprising *Prevotella copri*, *Bacteroides ovatus/cellulosilyticus/stercoris/eggerthii/xylanisolvens*, *Roseburia faecis*, *Eubacterium rectale*, *Coprococcus eutactus*, *Hungatella hathewayi*, *Dialister succinatiphilus/propionicifaciens*, *Bifidobacterium faecale/adolescentis*, *Lactobacillus*, *Pediococcus*, *Fusobacterium* and *Enterobacteriaceae* species (*De Paepe et al., 2017*; *De Paepe et al., 2018*).

An essential step in resolving the driving force and mechanisms behind this specific substrate attachment entails the study of pure cultures. To this end, species deposited in culture collections can be used, offering the advantage of working with fully characterized bacteria, from which genome information is available. Alternatively, bacteria can be isolated, permitting the discovery of novel strains and ensuring the use of relevant strains (*Greub, 2012*). Recent technological advancements have revived interest in bacterial culturing. The so-called culturomics approach, which involves high-throughput microbial culturing using different conditions and media, has shown that a large fraction of the gut microbial community is culturable (*Browne et al., 2016*; *Hugon et al., 2015*; *Lagier et al., 2012*; *Lagier et al., 2016*). While the automated picking and identification of millions of colonies is a promising strategy to capture microbial diversity in the gut, it is not a standard analysis that requires specialized equipment. As we are specifically interested in a subset of bacteria, capable of colonizing and metabolizing the wheat bran residue, we can considerably bring down our isolation efforts by targeted enrichment prior to isolation. To this end, in the work presented here, the fecal microbial communities derived from four healthy individuals were cultured using wheat bran as the sole nutrient source. The washed wheat bran residue with the attached microbiota, was subcultured four times in fresh medium to selectively enrich the wheat bran-colonizing and metabolizing species. Additionally,

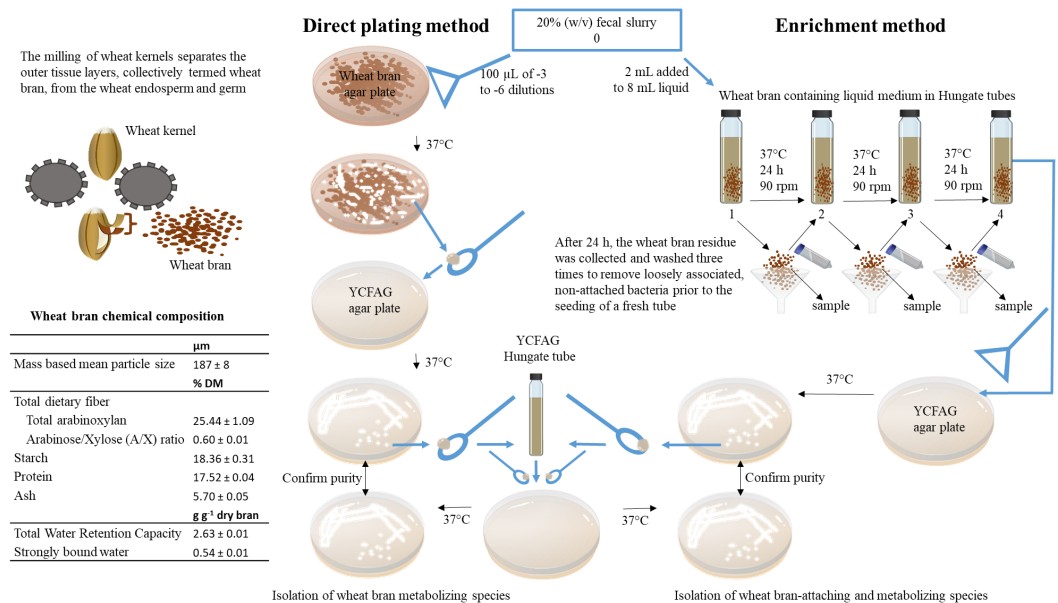

**Figure 1 Overview of the experimental set-up.** In this study, wheat bran-metabolizing and colonizing species were isolated from human fecal microbiota obtained from four healthy individuals. To this end, insoluble wheat bran particles with a characterized chemical composition were incorporated in solid agar plates used in a direct plating method on the one hand and added to a liquid broth used in an enrichment procedure on the other hand. As described in details in the materials and methods section of the manuscript, the fecal sample is directly plated onto the solid wheat bran agar plates to isolate wheat bran metabolizing species. Whereas, the enrichment method selects for wheat bran-attaching and metabolizing species by subculturing the wheat bran residue during three consecutive transfers.

the same fecal samples were directly plated on wheat bran agar, selecting only for wheat bran-metabolizing species.

# MATERIALS AND METHODS

Bacterial strains capable of metabolizing wheat bran as a sole nutrient source were isolated from a fecal slurry by a combination of direct plating and enrichment (Fig. 1). In order to account for inter-individual variability, the fecal sample of four different individuals was evaluated. Research incubation work with fecal microbiota from human origin was approved by the ethical committee of the Ghent University hospital under registration number B670201214538. Written informed consent was obtained from all participants. A fecal slurry was prepared according to *De Boever, Deplancke & Verstraete (2000)* and consisted of a 20% (w/v) fecal sample, suspended and homogenized in 0.1 M phosphate buffer pH 6.8, supplemented with 1 g $L^{-1}$ sodium thioglycolate (Sigma-Aldrich, St. Louis, MO, USA), henceforth referred to as 'anaerobic phosphate buffer' (*De Boever, Deplancke & Verstraete, 2000*). All isolation work was subsequently carried out in an anaerobic workstation (GP-Campus, Jacomex, TCPS NV, Rotselaar, Belgium).

## Solutions and growth media

All chemicals were purchased from Sigma-Aldrich (St. Louis, MO, USA). The isolation medium used in the direct plating and enrichment method, contained pre-digested particle size reduced wheat bran (50 g $L^{-1}$), with chemical composition determined as described in the supplementary information and displayed in Fig. 1. The medium was reinforced with vitamin and hemin stock solutions (Tables S1 and S2) and buffered at pH 5.8 and 6.8 respectively, to mimic proximal and distal colon pH using a 0.1 M phosphate buffer ($KH_2PO_4/Na_2HPO_4$) (*Cummings et al., 1987*; *Duncan et al., 2009*). The amount of wheat bran was reduced to 10 g $L^{-1}$ for donors 3 and 4.

For the direct plating method, the particle size reduced wheat bran was incorporated in the buffered autoclaved molten agar (15 g $L^{-1}$; Thermo Fisher Scientific, San José, CA, USA) medium prior to its solidification. Plates were poured in a laminar flow cabinet after addition of the filter sterilized heat labile stock solution (Table S2) and the wheat bran and stored at 4 °C, for maximum two weeks.

The enrichment was performed in a buffered liquid broth. Besides the vitamin solutions, resazurin (1 mg $L^{-1}$) was added as a redox indicator. After verifying the pH, the broth was heated with intermittent shaking to remove dissolved oxygen until boiling, after which it was sparged with $N_2$ gas (30 min) while the medium was cooling down in ice water. The medium was subsequently dispensed in Hungate tubes (Glasgeratebau Ochs Gmbh, Bovenden-Lenglern, Finland) (8 mL per tube) under a continuous gas flow and tubes were quickly sealed with butyl rubber stoppers and aluminum crimp seals to limit oxygen exposure. The headspace was flushed with $N_2$ for 30 cycles using a gas exchange apparatus (*Plugge, 2005*). The Hungate tubes were autoclaved. The filter sterilized heat labile stock solution (Table S2) and wheat bran were added in the anaerobic workstation, alongside 0.5 mL reducing reagent stock solution at the targeted pH (Table S3).

Purified, isolated bacterial colonies were further characterized in a defined general nutritional YCFAG medium (*Browne et al., 2016*; *Duncan et al., 2002*). The medium was modified by adding 0.1 M $KH_2PO_4/Na_2HPO_4$ and correcting the pH to 6.8 and 5.8 using 10 M NaOH and by replacing the cysteine-HCl and bicarbonate by the reducing reagent stock solution shown in Table S3. As for the isolation medium, this reducing reagent stock solution was added to the YCFAG broth right before use. The heat labile vitamins were added after autoclaving. YCFAG plates were poured in the laminar flow cabinet and stored at 4 °C. Anaerobic YCFAG liquid broth was prepared by boiling, sparging and flushing the medium as described before, except that 10 mL of the medium was distributed per tube. After subculturing to confirm purity, the obtained isolates in YCFAG medium were stored in cryovials at −80 °C in the presence of a cryoprotective agent (Table S4).

## Direct plating

In the anaerobic workstation, tenfold serial dilutions ($10^{-1}$ to $10^{-6}$) of the 20% (w/v) fecal slurry were prepared in 0.1 M anaerobic phosphate buffer at pH 6.8. The -3 to -6 dilutions were spread (100 µL) on the surface of a solid agar plate (both at pH 5.8 and 6.8) containing wheat bran as the sole nutrient source, using disposable sterile cell spreaders (VWR, Radnor, PA, USA). Plates were transferred to the anaerobic workstation at least 12 h

before inoculation. After inoculation, plates were incubated in the anaerobic workstation at 37 °C. To check for contamination, two plates, one for each pH, were inoculated with the anaerobic phosphate buffer used to prepare the serial dilutions, without the addition of a bacterial mix derived from the fecal slurry. Plates were daily inspected for growth. In case of perceivable growth, for each pH, ten discrete colonies were picked with an inoculating loop and streak plates were made on solid YCFAG agar medium. Plates were incubated in the anaerobic workstation at 37 °C. Again, YCFAG plates without bacterial suspension were included as a control. Growth on the YCFAG plates was monitored and if pure colonies were observed, a single colony was transferred to YCFAG broth in a Hungate tube, which was consequently incubated at 37 °C on an orbital shaker (90 rpm) in a 45° tilted position outside of the anaerobic working station. When visual growth occurred, 1 mL suspension was sampled and stored at −80 °C in the presence of a cryoprotective agent (1 mL) (Table S4) and as a control to confirm purity, an inoculating loop of suspension was subcultured on YCFAG plates, to visually assess conformity of the colony appearance.

## Enrichment

In the anaerobic workstation (GP-Campus, Jacomex, TCPS NV, Rotselaar, Belgium), 2 mL of the 20% (w/v) fecal slurry was inoculated in an enrichment tube, one for each pH, containing 8 mL isolation medium with wheat bran as the sole nutrient source. Two tubes without fecal inoculum were included as a control. The Hungate tubes were capped with butyl rubber stoppers and aluminum crimp seals and removed from the anaerobic workstation for incubation at 37 °C on an orbital shaker (90 rpm) in a 45° tilted position. After 24 h the Hungate tubes, including the controls, were transferred to the anaerobic workstation. The wheat bran residue was harvested on an autoclaved filter paper inserted in an autoclaved glass funnel and rinsed three times with anaerobic phosphate buffer to remove loosely attached luminal bacteria. The wheat bran residue was sampled with a disposable inoculating loop and five (donors 1 and 2) or two (donors 3 and 4) loops were transferred to a new Hungate tube with fresh isolation medium. This procedure was repeated three times. The supernatant of the last transfer (referred to as 'luminal suspension') was serially diluted ($10^{-1}$ to $10^{-6}$) in anaerobic phosphate buffer in a 96-well plate and the −2 to −6 dilutions were plated onto solid YCFAG agar medium. Single colonies (10 per pH) were isolated and pure cultures were obtained as described for the direct plating method.

## Sampling and analysis

The fecal slurry was aliquoted for the purpose of Short Chain Fatty Acids (SCFA) analysis and DNA extraction, followed by next-generation 16S rRNA gene amplicon sequencing. Colonies on the surface of the wheat bran containing solid agar plates were enumerated. The wheat bran residue and liquid broth in the enrichment Hungate tubes were sampled after each transfer for DNA extraction and SCFA analysis. Samples after the first and final enrichment from the low and high pH incubation for each donor were sent for next-generation 16S rRNA gene amplicon sequencing. For one donor, the complete sequence of enrichments was analyzed. The pure cultures in the YCFAG medium resulting

from both approaches were identified by 16S rRNA gene Sanger sequencing after DNA extraction and metabolically characterized by SCFA measurement. All data visualization and processing was performed in R version 3.4.2 (2017-09-28) (*R Core Team, 2016*), unless stated otherwise. The R code is provided in Data S1 and S2 under the form of an RMarkdown file and the knitted PDF version. Raw SCFA and 16S rRNA gene amplicon sequencing data is included in the Data S3–S9. The 16S rRNA gene Sanger sequences of the isolates are supplied as a compressed folder (Sanger_isolates.zip). Additionally, Data S10–S22 comprise (i) user defined functions, (ii) mothur reports with the closest 16S rRNA gene Sanger reference for each OTU obtained by 16S rRNA gene amplicon sequencing, (iii) OTU sequences obtained by 16S rRNA gene next-generation amplicon sequencing in FASTA format and (iv) RDP taxonomic annotation of the 16S rRNA gene Sanger sequences of the isolates, which are all imported in the RMarkdown file.

All samples for functional analysis and for DNA extraction (the pellet obtained after centrifuging 250 μL sample at 5,000$g$ for 10 min or 0.250 g washed bran residue) were stored at −20 °C. Samples for SCFA analysis of the enrichment tubes and fecal slurry were 1:2 diluted in demineralized water prior to analysis.

SCFA analysis and a phenol-chloroform based DNA purification, following DNA extraction through chemical and mechanical lysis by multidirectional beating were performed according to *De Paepe et al. (2017)*. The DNA quality was verified by electrophoresis on a 1.5% (w/v) agarose gel and the DNA concentration was measured using the QuantiFluor® dsDNA kit (Promega, Madison, WI, USA) and Glomax®-Multi+ system (Promega, Madison, WI, USA).

The 16S rRNA gene from the pure cultures was amplified by PCR with the 63F (5′CAGGCCTAACACACATGCAAGTC3′)—1378R (5′CGGTGTGTACAAGGCCCGGG AACG3′) primer pair in a BioRad T100™ Thermal Cycler (Applied Biosystems, Foster City, CA, USA) (*Lane, 1991*). Primers were synthesized by Biolegio (Nijmegen, The Netherlands) and added in a final concentration of 0.2 μM in sterile nuclease-free water (Sigma-Aldrich, St. Louis, MO, USA), containing 0.1 μL Taq buffer μL$^{-1}$ PCR-mix, 0.025 units Recombinant Taq DNA polymerase μL$^{-1}$ PCR-mix, 0.2 μM dNTP Mix, 1.5 μM MgCl$_2$ (Fermentas Molecular Biology Tools, Waltham, MA, USA), 0.75 μM BSA (Roche Applied Science, Penzberg, Germany) and 0.04 μL DNA extract μL$^{-1}$ PCR-mix. The PCR amplification was initiated by a pre-denaturation step (5 min at 94 °C), followed by repeated denaturation (1 min at 95 °C), annealing (1 min at 53 °C) and extension (2 min at 72 °C) for 30 cycles, followed by 10 min at 72 °C. PCR-products were purified with the innuPREP PCRpure Kit (Analytik Jena, Jena, Germany) and sent for molecular identification by bi-directional Sanger sequencing (LGCGenomics, Teddington, Middlesex, UK). Forward and reverse 16S rRNA gene Sanger reads were classified through the RDP web interface using the RDP SeqMatch tool, restricting the database search to type strains with only near-full-length good quality sequences, and blasted in NCBI against the 16S rRNA gene sequences, selecting only type material, with optimization of the BLAST algorithm for highly similar sequences (accession date: June 2017) (*Altschul et al., 1990*; *Cole et al., 2014*; *Wang et al., 2007*). Results were manually compared and yielded a good correspondence. Bioedit was used to assess sequence quality, by manual inspection of the sequence traces

in the chromatograms (*Hall, 1999*). Short reads or reads with a lot of ambiguous base calls were precluded from the analysis.

Next-generation 16S rRNA gene amplicon sequencing of the V4 region (515F-806R) was performed on an Illumina MiSeq platform (Illumina, Hayward, CA, USA) using Illumina MiSeq v2 chemistry at the VIB Nucleomics core (VIB, Gasthuisberg Campus, Leuven, Belgium). Positive and negative controls were taken along as discussed in *De Paepe et al. (2018)*.

The mothur software package (v.1.39.5) and guidelines were used to process the amplicon data as described in detail in *De Paepe et al. (2018)* (*Kozich et al., 2013*). An OTU is hereinafter defined as a collection of sequences with a length between 220 and 253 nucleotides that are found to be more than 97% similar to one another in the V4 region of their 16S rRNA gene after applying OptiClust clustering (*Chen et al., 2013*; *Schloss & Westcott, 2011*; *Schloss et al., 2009*; *Wang et al., 2012*). Taxonomy was assigned using the RDP version 16 and silva.nr_v123 database (*Cole et al., 2014*; *Quast et al., 2013*; *Wang et al., 2007*). The resulting OTU table and taxonomy file were loaded in R (*R Core Team, 2016*). All samples from donor 4, except for the final enrichment step, were discarded due to an insufficient number of reads (<100). The outcome of the enrichment procedure was assessed by computing richness (Chao1 Richness estimator) and diversity (Shannon, Simpson, inverse Simpson and Fisher alpha) estimators using vegan_2.4-4 (*Oksanen et al., 2016*). The proportional community composition was displayed in bar graphs. For this purpose, the OTU table was filtered according to the arbitrary cutoff's described by *McMurdie & Holmes (2014)*, whereby OTUs observed in less than 5% of the samples and with read counts below 0.5 times the number of samples were removed (*McMurdie & Holmes, 2014*). At genus level RDP version 16 taxonomy is displayed. To arrive at a species level classification, OTUs were manually annotated using the RDP web interface using the RDP SeqMatch tool, restricting the database search to type strains with only near-full-length good quality sequences, and blasted in NCBI against the 16S rRNA gene sequences, selecting only type material, with optimization of the BLAST algorithm for highly similar sequences (accession date: June 2017) (*Altschul et al., 1990*; *Wang et al., 2007*; *Cole et al., 2014*). Inconsistent species level taxonomy assignments were not reported. The sequence data has been submitted to the NCBI database under accession number SRP091975.

Finally, the resulting OTUs were compared to the 16S rRNA gene Sanger reads of the obtained isolates by means of a phylogenetic placement analysis. Forward and reverse Sanger sequences for each donor were grouped into separate files and reverse complements and summary statistics were obtained using the mothur software package (v.1.39.5) (*Schloss et al., 2009*). The 515F-806R primer pair, used for Illumina MiSeq 16S rRNA gene amplicon sequencing was located in the Sanger reads (forward and reverse compliment). In case both primers were not present on one and the same read (either forward, or reverse), consensus sequences (contigs) were generated using the sangeranalyseR package (version 0.1.0) (*Lanfear, 2015*). Contigs with more than 100 degenerated positions, indicative of a poor quality alignment, were omitted. A reference alignment was built from the Sanger reads (for each donor separately) applying the sina aligner (*Pruesse, Peplies & Glockner, 2012*).

OTUs were aligned to this reference Sanger alignment in mothur (align.seqs), yielding a report with the closest Sanger reference for each OTU based on kmer searching (*Schloss et al., 2009*). This report was loaded into R (*R Core Team, 2016*). For each isolate in the report, the top two OTUs with the highest SearchScores were selected. A FASTA file was constructed containing these OTUs. In order to compare the OTUs spanning the V4 region of the 16S rRNA gene with the near full-length Sanger reads, the RAxML implementation of the evolutionary placement algorithm of short reads, as introduced by *Berger, Krompass & Stamatakis (2011)*, was used (*Stamatakis, 2014*). The bootstrap supported maximum likelihood (ML) phylogenetic reference tree was also constructed using RAxML, selecting the General Time Reversible model of nucleotide substitution under the Gamma model of rate heterogeneity (GTRGAMMA) with the parsimony random seed set to 12345. The rapid bootstrap analysis was conducted starting from $N = 1,000$ distinct randomized maximum parsimony trees and was followed by a search for the best-scoring ML tree with rapid bootstrap random number seed 123 (*Stamatakis, 2014*). The best scoring ML tree with the OTU short read insertions was visualized in iTOL (*Letunic & Bork, 2016*). The proportional abundance of the OTUs in the fecal slurry and in the luminal suspension after the last passage at pH 5.8 and 6.8 were integrated in the tree as a multi-value bar chart.

### SEM and cryo-SEM visualization

Native, pre-digested and fermented wheat bran samples were visualized using cryo-SEM and desktop SEM. For the purpose of SEM microscopy, the bran samples were chemically dried with hexamethyldisilazane (HMDS) as described by *Araujo et al. (2003)*. After complete evaporation of the HMDS, samples were mounted on an aluminum pin (diameter: 12 mm) using double sided carbon tape and subsequently gold sputtered for 45 s at 30 mA (Agar Sputter Coater B7340, Agar Scientific, UK). Images were collected using a Phenom Pro X SEM microscope (Phenom-World B.V., the Netherlands) with a beam intensity of 10 keV.

As an alternative to SEM microscopy, samples were also visualized via cryo-SEM using a Jeol JSM 7100F scanning electron microscope (JEOL Ltd, Tokyo, Japan). A small amount of wheat bran was placed on a sticky carbon surface mounted on an aluminium stub, vitrified in a nitrogen slush and transferred under vacuum conditions into the cryo-preparation chamber (PP3010T Cryo-SEM Preparation System; Quorum Technologies, Lewes, UK) conditioned at −140 °C. Subsequently, the sample was sublimated for 20 min at −70 °C to remove frost artefacts, sputter-coated with platinum using argon gas, transferred to the SEM stage at −140 °C and electron beam targeted at 3 keV.

## RESULTS & DISCUSSION

In an attempt to isolate human fecal bacteria capable of growing on and attaching to wheat bran, two different approaches were adopted: direct plating on a wheat bran based solid agar medium and plating the cultures after a series of enrichment steps. Inter-individual variability and pH were previously shown to determine the outcome of wheat bran colonization (*Cummings et al., 1987*; *Duncan et al., 2009*) and were accounted for in the present study by examining four different donors at two pH values representative for proximal (pH 5.8) and distal (pH 6.8) colon conditions. As wheat bran was not autoclaved

to avoid structural modifications, a control wheat bran sample was incubated under both pH conditions. Bacterial growth was observed in the control, despite the pre-incubation at low pH in the presence of digestive enzymes, mimicking the gastro-intestinal transit. 16S rRNA gene sequencing of the V4 region, however, revealed that one OTU accounted for 100% of the reads. This *Enterobacteriaceae* OTU could not be unambiguously classified and was not recovered in any of the incubations inoculated with a fecal sample, nor did anything grow on the YCFAG plates after enrichment of the control.

## Enrichment of wheat bran-attached bacteria

Incubation of fecal microbiota with wheat bran as a sole nutrient source resulted in short chain fatty acid (SCFA) production, concomitant with a marked acidification, confirming the microbial growth that was visually observed as an increased optical density (Fig. 2). During the consecutive enrichment steps, the microbial richness and diversity decreased, as illustrated in the case of donor 1 (Fig. 3), confirming the effectiveness of the enrichment procedure. After the first 24 h incubation of the fecal sample of donor 1 in the presence of wheat bran, the bran-attached community was clearly enriched in *Bifidobacterium* OTU1 (most similar to *B. faecale/adolescentis*) at pH 5.8 and *Dialister* OTU12 at pH 6.8 (Fig. 4). This wheat bran residue served as an inoculum for the next 24 h incubation, a procedure which was repeated three times in total. The enrichment of bifidobacteria at low pH persisted during successive passages and lactobacilli appeared from the second enrichment step onwards. Acetate was the main metabolic end product detected, but it must be noted that lactate production was not measured. In the pH 6.8 condition, butyrate was formed next to acetate, which can be linked to the enrichment of *Faecalibacterium prausnitzii* OTU5 besides *Bifidobacterium* OTU1. In the other two donors, next to *Lactobacillus*, bifidobacteria and *F. prausnitzii*, an enrichment of *Pediococcus*, *Enterobacteriaceae*, *Escherichia/Shigella* and *Fusobacterium* species was observed (Figs. 5 and 6). The successful colonization of bifidobacteria, *Lactobacillus* and *Pediococcus* on wheat bran has been observed in the proximal colon compartment of the Dietary Particle Mucus Simulator of the Human Intestinal Microbial Ecosystem (DP-M-SHIME) (*De Paepe et al., 2018*). The efficient wheat bran colonization could possibly be attributed to extensive adhesive properties and might involve the expression of pili or the production of EPS (*Foroni et al., 2011*; *Hidalgo-Cantabrana et al., 2014*; *Johansson et al., 2011*; *Kankainen et al., 2009*; *Lebeer et al., 2012*; *Roos & Jonsson, 2002*; *Ruas-Madiedo et al., 2007*; *Van den Abbeele et al., 2013*; *Van den Abbeele et al., 2009*; *Van den Abbeele et al., 2012*). Their preference for the proximal colon compartment was suggested to reflect their acid tolerance (*Chung et al., 2016*; *De Paepe et al., 2018*; *Duncan et al., 2009*; *Tannock, 2004*; *Van den Abbeele et al., 2012*; *Walker et al., 2005*). In line with this, *Lactobacillus* and *Pediococcus* were confined to the pH 5.8 enrichments in donors 1 and 2, which were characterized by a large pH drop by more than one pH unit (Fig. 2). In order to reduce the degree of acidification and associated enrichment of pH tolerant species, the wheat bran concentration and seeding amount were lowered from 50 to 10 g L$^{-1}$ and from five to two inoculating loops respectively in donors 3 and 4. This resulted in a more moderate pH decrease to $6.36 \pm 0.15$ and pH $5.18 \pm 0.19$ and reduced the share of lactobacilli in the enrichments. Donor 3 and 4, were instead

characterized by large *Escherichia/Shigella* proportions, both at pH 5.8 and 6.8. Adhesive structures are well documented in *Enterobacteriaceae* species and colonization of the wheat bran residue might bestow resilience to a suboptimal lower pH (*Duncan et al., 2009*; *Gaastra et al., 2014*; *Lawley & Walker, 2013*; *The et al., 2016*). *Enterobacteriaceae* were less present on the bran in the DP-M-SHIME and mostly resided in the mucus layer. In the absence of a mucus micro-environment in static batch incubations, however, we previously found *Enterobacteriaceae* to dominate the early stages of wheat bran colonization (*De Paepe et al., 2018*, submitted to ISME Journal). The latter was also observed for *Fusobacteria*, which were abundant in the final enrichment stage of donor 2 in the present experiment. Finally, the establishment of *F. prausnitzii* in donors 1, 3 and 4 is surprising and was not perceived in any of the previous experiments. *F. prausnitzii* is positioned in the mucus layer, close to the gut epithelium and is able to adhere to mucins, but not to epithelial cells (*Khan et al., 2012*; *Martín et al., 2017*; *Marzorati et al., 2014*). Moreover *F. prausnitzii* was shown to adhere to wheat bran in a synbiotic formulation (*Khan, van Dijl & Harmsen, 2014*). Little is known regarding its mechanism of adhesion.

Besides the capacity to adhere, the enriched bacteria depended on wheat bran as the sole nutrient source. In general, micro-organisms need a carbon source as building blocks for organic matter, a nitrogen/phosphorous/sulphur source for protein and nucleic acid synthesis, some trace elements such as iron, magnesium, cobalt and manganese as cofactors for enzymes, and, in case of auxotrophic growth, some vitamins and amino acids (*Madigan & Brock, 2011*). Wheat bran is a versatile substrate, containing all of the above compounds (*Hemdane et al., 2016*). The compounds are, however, part of a complex macro-molecular configuration. Carbohydrates in wheat bran mainly consist of non-starch cell wall polysaccharides (NSP), comprising arabinoxylans, β-glucan, cellulose and fructan, which are physically intertwined with lignin (*Hemdane et al., 2016*). Some residual starch can be present in the attached endosperm resulting from the crude milling process (Figs. 7 and 8) (*Hemdane et al., 2016*; *United States Department of Agriculture (USDA), 2016*). The residual endosperm fraction also contains some proteins (*Chick et al., 1947*; *Shewry, 2009*). The major part of the endosperm components is expected to disappear during a pre-digestion with gastric and pancreatic enzymes (Fig. 7) (*Amrein et al., 2003*; *Chick et al., 1947*). Wheat bran proteins, however, are also located in aleurone cells, which are more recalcitrant to digestion (*Amrein et al., 2003*; *Arte et al., 2015*). This complex wheat bran structure and its insoluble nature limits fermentability. In that sense, the retrieved enriched bacteria are unexpected as none of the species is considered to be a wheat bran primary degrader, judged by the fact that they are not equipped with the enzymatic complement required to solubilize and degrade cellulose and arabinoxylan polymers.

Bifidobacteria generally prefer arabinoxylan oligosaccharides (AXOS) as a substrate (*Van den Broek et al., 2008*). *Bifidobacterium longum* and *Bifidobacterium adolescentis*, however, are able to grow on arabinoxylans (*Savard & Roy, 2009*; *Van den Broek et al., 2008*). Genomic studies confirmed the presence of arabinofuranosidases and xylosidases in these species. *Bifidobacterium longum* also possesses a multi-domain enzyme with a putative endo-xylanase (GH43) flanked by two carbohydrate-binding modules (CBM) that might interact with xylans. (*Van den Broek et al., 2008*). Arabinoxylan or AXOS consumption by

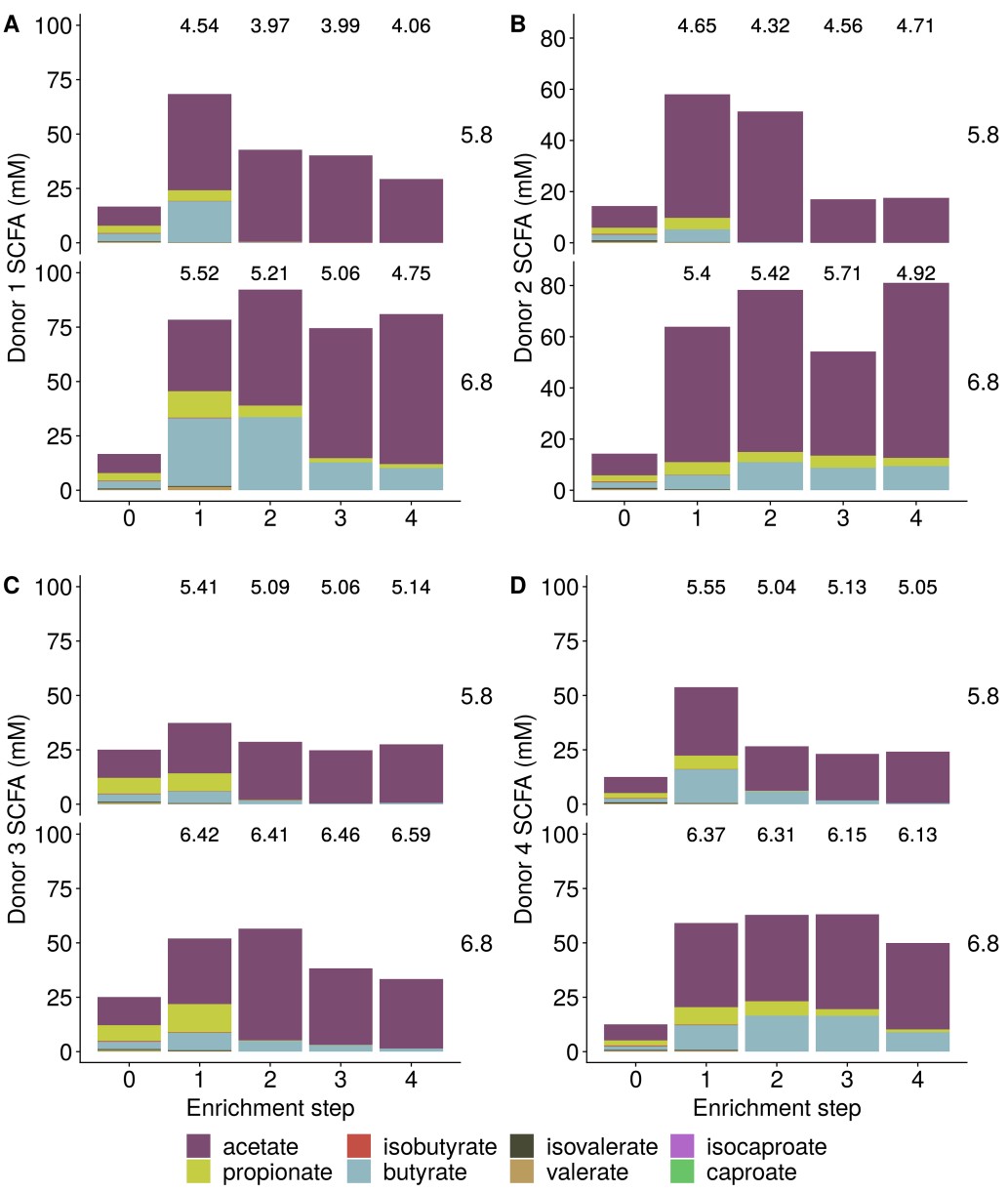

**Figure 2   Short Chain Fatty Acid (SCFA) production by the fecal microbiota derived from four different donors (A–D) during enrichment with wheat bran as the sole nutrient source.** The fecal sample (enrichment step 0) was incubated with wheat bran for 24 h (enrichment step 1), after which the wheat bran residue was washed to remove loosely attached bacteria and used to seed a new incubation (enrichment step 2). This procedure was repeated two more times (enrichment step 3 and 4). The pH is indicated on top of the stacked bars and decreased considerably compared to the starting pH (5.8 and 6.8).

lactic acid bacteria (LAB) received less attention, so far (*Michlmayr et al., 2013*). Lactobacilli have been reported to respond to AXOS treatment in the SHIME model but *Lactobacillus brevis* is the only species for which arabinofuranosidase and xylosidase activity is evidenced (*Michlmayr et al., 2013*; *Sanchez et al., 2009*). Based on functional predictions, it is suggested

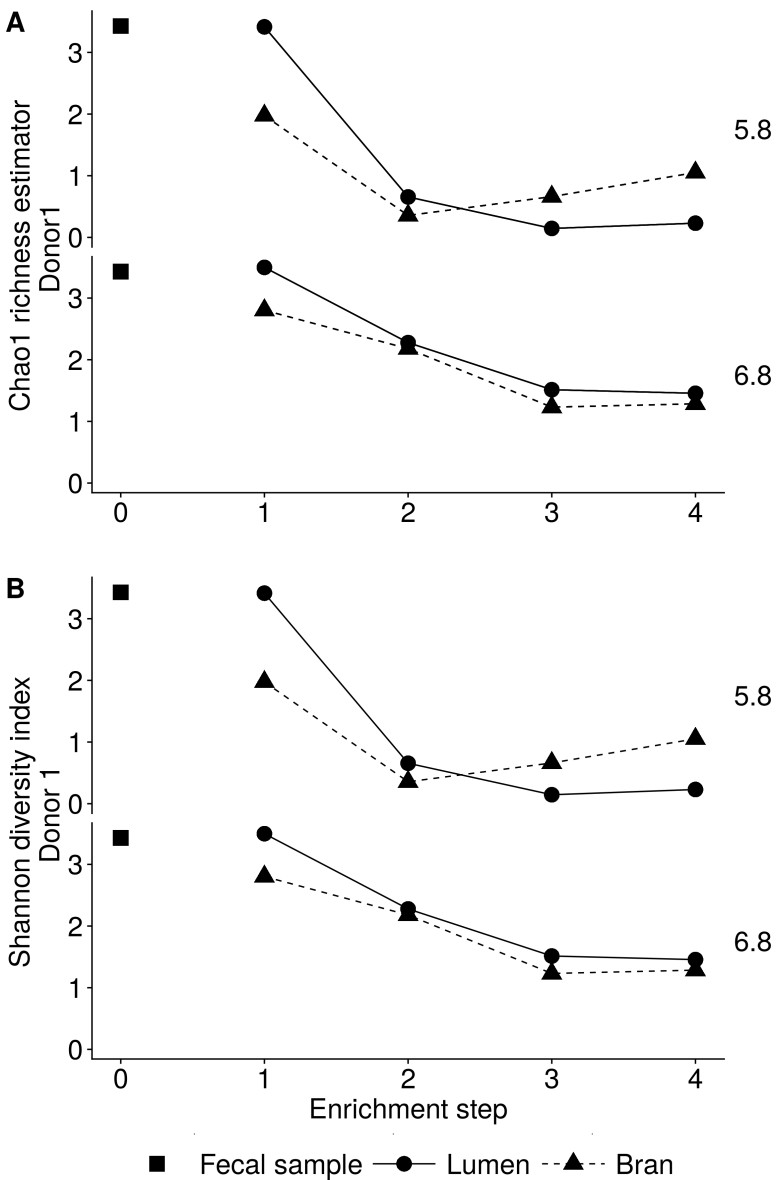

**Figure 3   Reduction in the microbial community richness (A, Chao 1 richness estimator) and diversity (B, Shannon diversity index) during consecutive enrichment steps with wheat bran as the sole nutrient source, as shown for donor 1.** The fecal sample (enrichment step 0) was incubated with wheat bran for 24 h (enrichment step 1), after which the wheat bran residue was washed to remove loosely attached bacteria and used to seed a new incubation (enrichment step 2). This procedure was repeated two more times (enrichment step 3 and 4).

that the AXOS degrading capacity might be shared by *Pediococcus* and *Enterococcus* spp. (*Michlmayr et al., 2013*). Interestingly, despite this apparently limited wheat bran degrading capacity, our findings are consistent with *Enterococcus* spp., *Lactobacillus* spp. and *Pediococcus pentosaceus* being the main species recovered from spontaneous bran fermentations and the fact that wheat bran is often used as a substrate in industrial

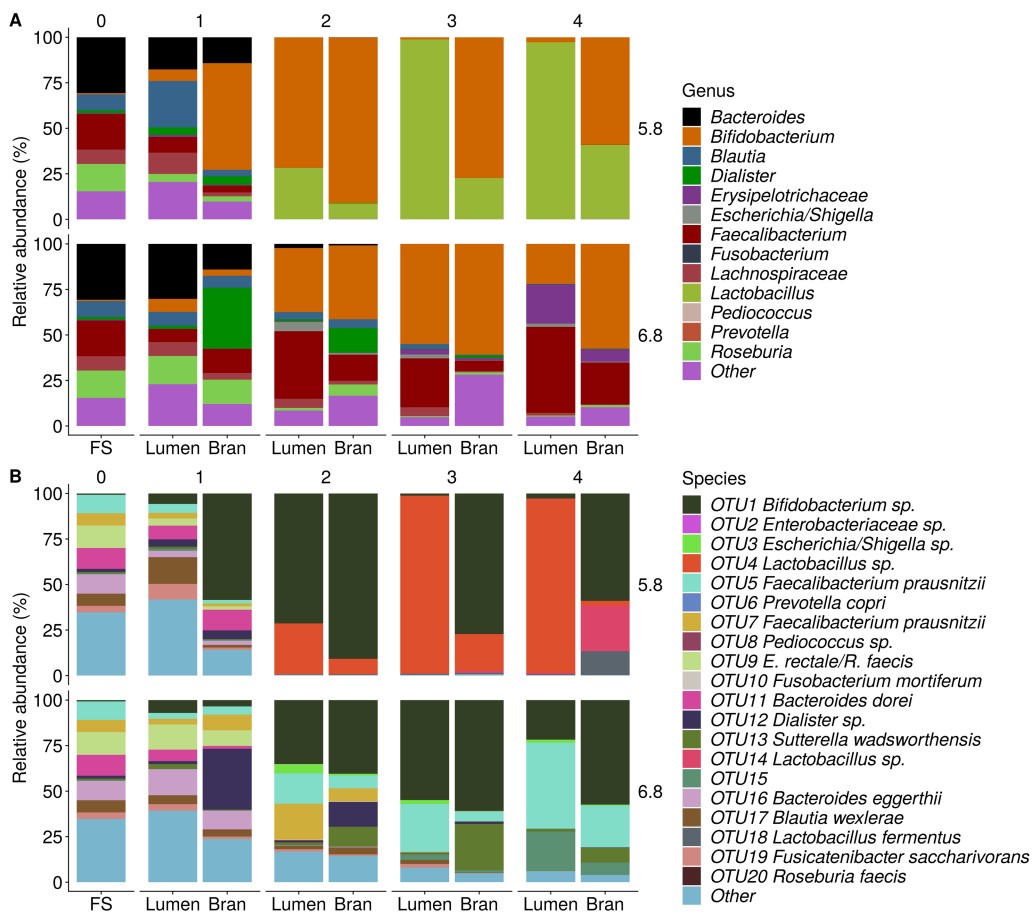

**Figure 4** **Shifts in genus (A) and species (B) level microbial community composition of donor 1 during consecutive enrichment steps with wheat bran as the sole nutrient source.** The fecal sample (FS; enrichment step 0) was incubated with wheat bran for 24 h (enrichment step 1), after which the wheat bran residue was washed to remove loosely attached bacteria and used to seed a new incubation (enrichment step 2). This procedure was repeated two more times (enrichment step 3 and 4). Family level taxa appearing in the genus level plots should be interpreted as unclassified genus belonging to the respective family.

fermentations with lactobacilli (*Arte et al., 2015*; *Katina et al., 2012*; *Prückler et al., 2015*). In this context, native wheat bran is used, containing more residual starch. Moreover, the activity of endogenous wheat bran endoxylanases and cinnamoyl esterases, solubilizes the arabinoxylans and increases the availability of arabinose and xylose monomers, which can be used by lactobacilli (*Katina et al., 2012*; *Prückler et al., 2015*). Besides endogenous xylanases, wheat kernels can contain xylanases from microbial origin (*Dornez et al., 2006*). The fate of these wheat bran associated xylanases and other endogenous wheat bran polymer degrading enzymes during *in vivo* gastro-intestinal digestion is unknown. But, a reduced, yet, preserved activity of those enzymes has been observed after *in vitro* pre-digestion (K De Paepe, 2015, unpublished data), which could explain the isolation of species lacking these first-line enzymes. *Enterobacteriaceae* and *Fusobacterium* species are capable of saccharolytic fermentation, but are also lacking xylan and cellulose degrading enzymes

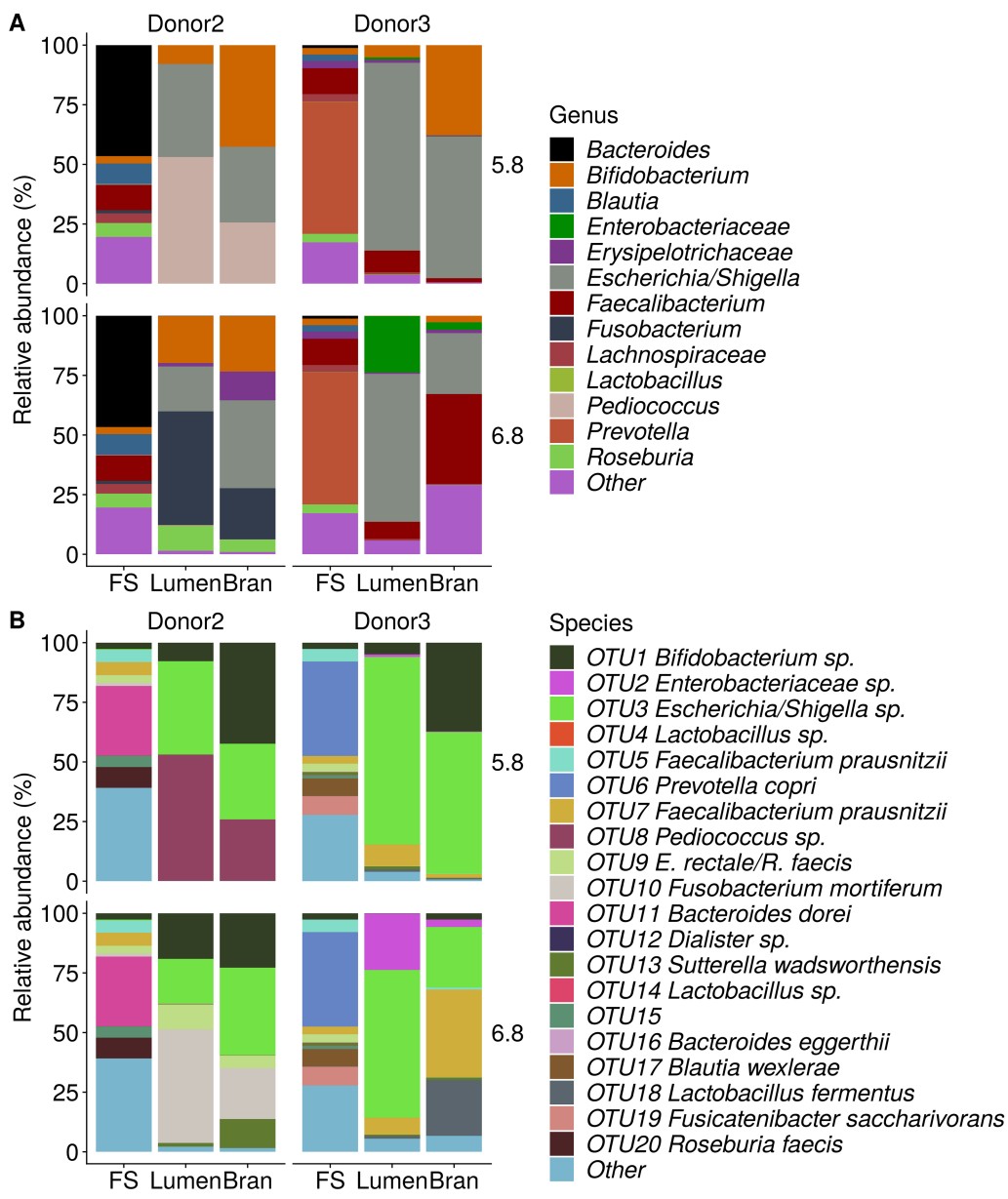

**Figure 5** **Shifts in genus (A) and species (B) level microbial community composition of donor 2 and 3 after four enrichment steps with wheat bran as the sole nutrient source.** The fecal sample (FS; enrichment step 0) was incubated with wheat bran for 24 h (enrichment step 1), after which the wheat bran residue was washed to remove loosely attached bacteria and used to seed a new incubation (enrichment step 2). This procedure was repeated two more times (enrichment step 3 and 4). Only the final enrichment step and FS are shown in this plot. Family level taxa appearing in the genus level plots should be interpreted as unclassified genus belonging to the respective family.

(*Brady et al., 2009*; *Cantarel et al., 2009*; *Gao et al., 2015*; *Lombard et al., 2014*; *Marzorati et al., 2017*; *Mazur & Zimmer, 2011*; *Richardson, McKain & Wallace, 2013*; *Robrish, Oliver & Thompson, 1991*; *Salamanca-Cardona et al., 2014*; *Thompson et al., 1997*). *F. prausnitzii* is shown to degrade wheat bran in pure cultures to a small extent (~4% substrate loss)

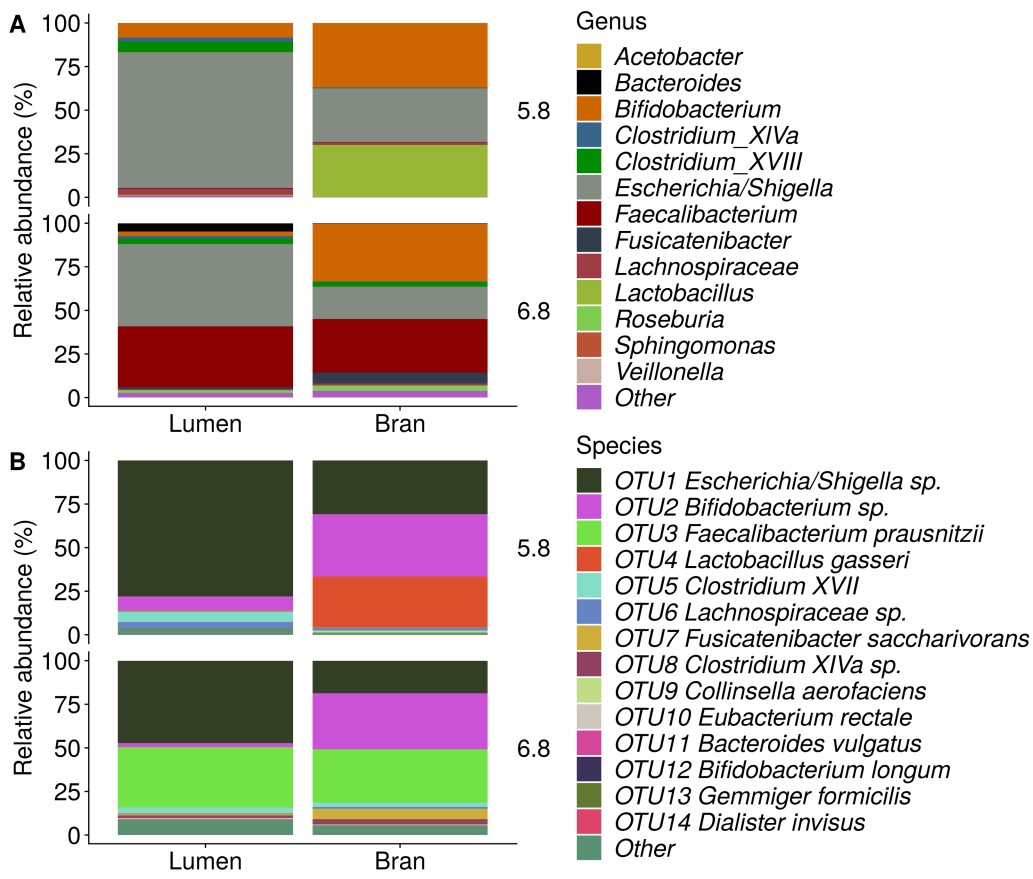

**Figure 6** **Genus (A) and species (B) level microbial community composition of donor 4 after the final enrichment step with wheat bran as the sole nutrient source.** The fecal sample (FS; enrichment step 0) was incubated with wheat bran for 24 h (enrichment step 1), after which the wheat bran residue was washed to remove loosely attached bacteria and used to seed a new incubation (enrichment step 2). This procedure was repeated two more times (enrichment step 3 and 4). Only the final enrichment step was shown in this plot, due to a low read number (< 100 reads) in the fecal sample of donor 4. Family level taxa appearing in the genus level plots should be interpreted as unclassified genus belonging to the respective family.

(*Duncan et al., 2016*). The required enzymatic activity is not experimentally characterized but genomic predictions have identified several glycosyl hydrolases which can display β-xylosidase (EC 3.2.1.37), α-L-arabinofuranosidase (EC 3.2.1.55), endo-1,4-β-xylanase (EC 3.2.1.8) and feruloyl esterase (EC 3.1.1.73 ) activity (GH1,GH3,GH31,GH43,CE1), as well as a possible xylan binding domain (CBM13) (*Cantarel et al., 2009*; *Lombard et al., 2014*). The release of ferulic acid from wheat bran by a pure culture of *F. prausnitzii* has been demonstrated. Additionally, *F. prausnitzii* can benefit from the high acetate concentrations through cross-feeding, supporting our finding of *F. prausnitzii* to be one of the isolates (*Duncan et al., 2002*; *Rios-Covian et al., 2015*).

Besides the carbohydrate degrading capacity, growth on wheat bran as the sole nutrient source requires proteolytic activity. The breakdown of proteins is essential for the assimilation of peptides and amino-acids into new microbial biomass and can sustain

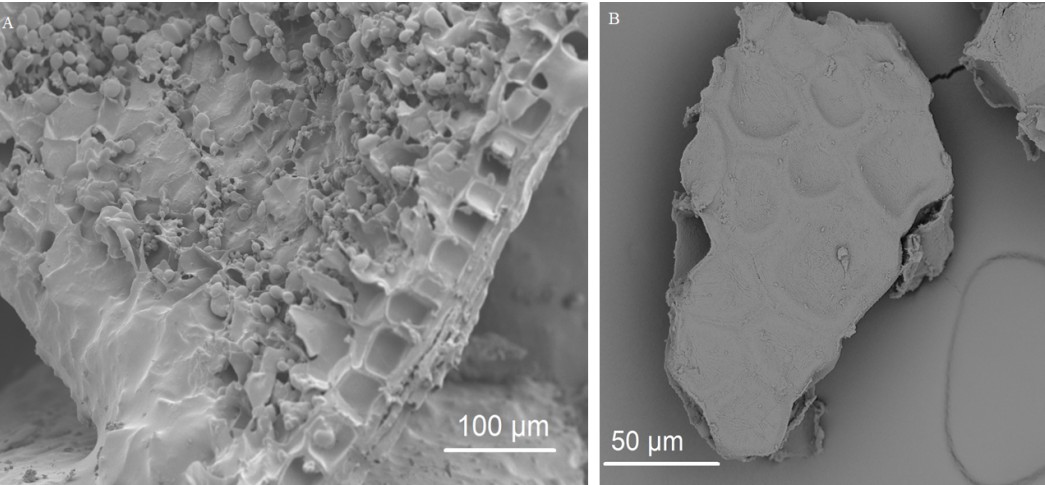

**Figure 7** **Removal of starch during pre-digestion.** (A) Cryo-SEM image of an unmodified wheat bran particle with endosperm starch granules, resulting from the crude milling process, covering the surface of aleurone cells. (B) SEM image of a micronized wheat bran fragment after pre-digestion, without attached starch granules.

fermentation leading to the production of branched SCFA and ammonium (*Macfarlane, Cummings & Allison, 1986*). Ammonium was not measured in this study, and branched SCFA concentrations were negligible, despite the proteolytic fermentation capacity of some of the enriched species (*Enterobacteriaceae, Fusobacterium*) (*Marzorati et al., 2017*; *Richardson, McKain & Wallace, 2013*; *Robrish, Oliver & Thompson, 1991*). The breakdown of wheat bran proteins by gut bacteria is poorly studied. Wheat bran fermentation by LAB starter cultures has indicated that enzymes from microbial origin contribute to the size reduction of oligopeptides and the generation of free amino acids, while the release of oligopeptides depended on the endogenous wheat bran proteases (*Arte et al., 2015*). Isolation experiments with gluten proteins revealed that *Enterococcus, Bifidobacterium, Pediococcus, Lactobacillus* and *Bacteroides fragilis* are capable of hydrolyzing gluten proteins and derived peptides (*Caminero et al., 2014*). Interestingly, *F. prausnitzii* and *Bacteroides dorei* were recovered from liquid incubations with gluten proteins as the main nitrogen source, but could not be isolated from solid agar media (*Caminero et al., 2014*).

## Isolation of wheat bran degrading and attached bacteria

The luminal suspension obtained after the final enrichment step was plated onto nutritional medium and alternatively, the fecal sample was directly plated on wheat bran based solid agar plates. Control wheat bran agar plates, which were inoculated with anaerobic phosphate buffer, without the addition of a fecal sample showed no growth. From both strategies, at pH 5.8 and 6.8, ten single colony isolates were picked. Isolates were identified through 16S rRNA gene Sanger sequencing (Table 1).

Plating of the enrichment suspension resulted in the isolation of *Lactobacillus* and *Collinsella* species in donor 1, *Escherichia/Shigella*, *Pediococcus*, *Bifidobacterium* and *Enterococcus* species in donor 2, *Escherichia/Shigella*, *Enterococcus* and *Klebsiella* species in

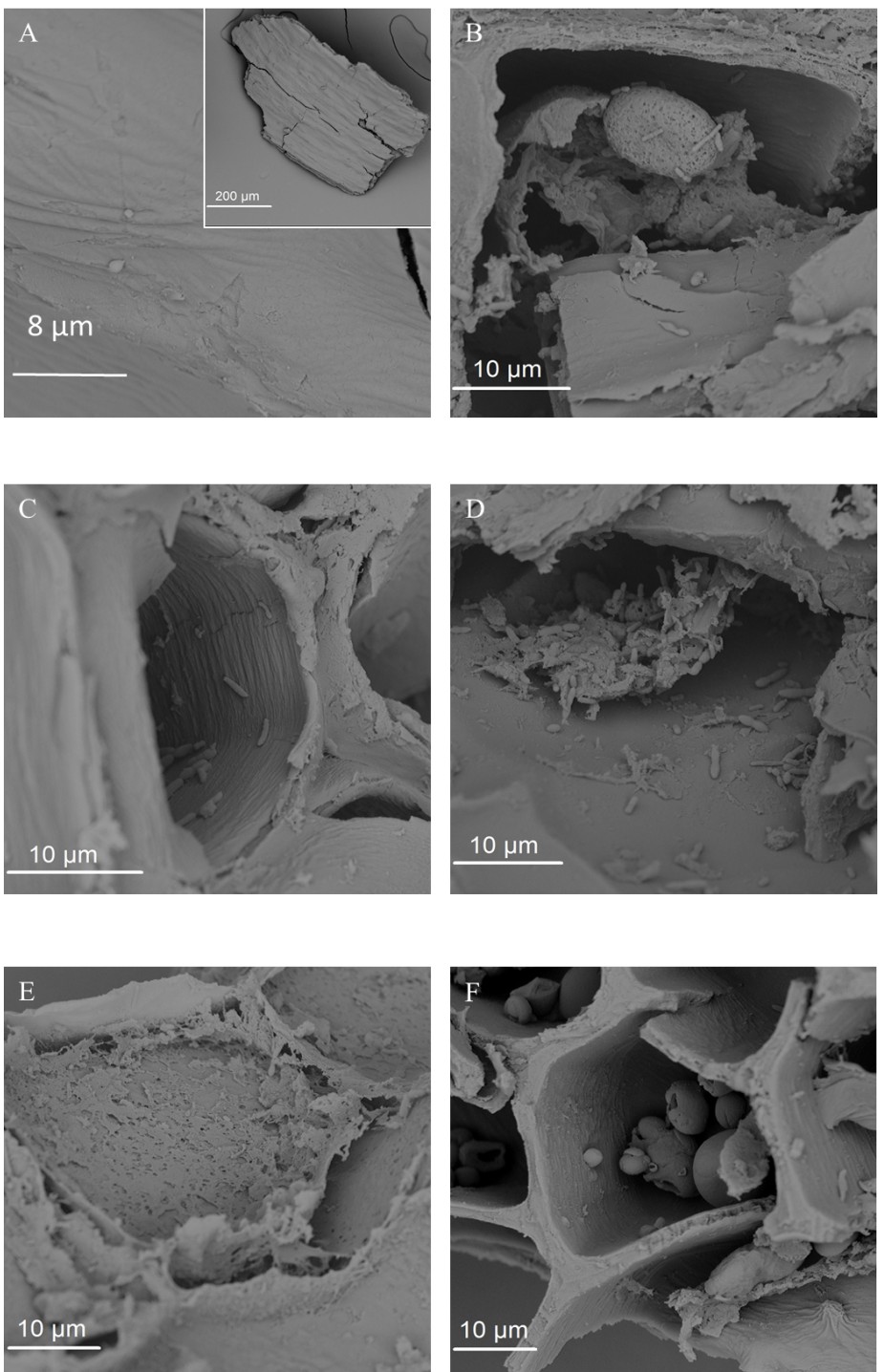

**Figure 8  SEM images of the fermented wheat bran residue after 24 h.** (A) Uncolonized pericarp tissue. (B) Residual starch granule. (C) Bacteria inside a wheat aleurone cell. (D) Bacteria on aleurone fragment. (E) Partial degradation of aleurone cell. (F) Ligature at the junction of aleurone cells.

**Table 1  RDP Seqmatch and NCBI BLAST results from the single colony isolates.** The best hits, with the highest percentage identity (NCBI) and similarity score (S_ab in RDP) are shown. Species obtained by direct plating are indicated in boldface.

| Isolate | Best NCBI/RDP match (>98% identity) |
| --- | --- |
| **Donor 1** | |
| 83,95 | *Bacillus anthracis/toyonensis/cereus/thuringiensis/ pseudomycoides/mycoides/weihenstephanensis/ marcorestinctum/bingmayongensis/manliponensis/ gaemokensis/cytotoxicus* |
| **78** | *Bacteroides eggerthii* |
| **84,87** | *Bacteroides dorei* |
| **79,85** | *Bacteroides fragilis* |
| **86** | *Bacteroides thetaiotaomicron/faecis* |
| **76** | *Bifidobacterium bifidum* |
| **80,88,90** | *Bifidobacterium adolescentis/faecale/ruminantium* |
| **82**,91,92,93,94,96,97,99 | *Collinsella aerofaciens* |
| **81** | *Dorea formicigenerans* |
| 103,105,106,107,108,109,110 | *Lactobacillus delbrueckii* |
| 101,102,104 | *Lactobacillus fermentum* |
| **Donor 2** | |
| **19,25,28** | *Bifidobacterium faecale/adolescentis* |
| **21** | *Bifidobacterium bifidum* |
| **26** | *Bifidobacterium pseudocatenulatum* |
| 34 | *Bifidobacterium faecale/adolescentis/ruminantium* |
| **10,27**,30,31,32,35,37 | *Escherichia coli/fergusonii/marmotae/vulneris/albertii* |
| **29** | *Shigella sonnei/flexneri/dysenteriae/boydii Enterococcus durans* |
| 36,40,41,42,43,44,45,46 | *Pediococcus pentosaceus* |
| **13,15,18** | *Streptococcus pasteurianus/macedonicus/equinus/ gallolyticus/lutetiensis/loxodontisalivarius/ infantarius/saliviloxodontae* |
| **Donor 3** | |
| 153 | *Enterococcus faecium* |
| **131,138**,146,151,152,153,154 ,156,157,158,159,160,161 | *Enterococcus faecium/lactis/durans/villorum/hirae/ thailandicus/mundtii/dispar/canintestini/ratti/raffinosus/ pseudoavium/ casseliflavus/avium/viikkiensis/gilvus/ malodoratus/devriesei/gallinarum/xiangfangensis/canis/*<br><br>*asini/massiliensis/pallens* |
| 143,144,149 | *Shigella sonnei/flexneri*<br><br>*Escherichia fergusonii* |
| **128,129**,141,145,147 | *Klebsiella michiganensis/oxytoca/pneumoniae/ quasipneumoniae, Enterobacter cloacae/bugandensis/ cancerogenus/asburiae/xiangfangensis*<br><br>*Escherichia vulneris, Yokenella regensburgei*<br><br>*Kluyvera cryocrescens*<br><br>*Raoultella ornithinolytica/terrigena* |

**Table 1** (*continued*)

| Isolate | Best NCBI/RDP match (>98% identity) |
| --- | --- |
| **133,137,140** | *Streptococcus lutetiensis/infantarius/equinus pasteurianus/macedonicus* |
| **136,**155 | *Pediococcus pentosaceus/stilesii/claussenii* |
| **Donor 4** | |
| **184,185** | *Bacteroides ovatus* (90%,96% similarity) |
| **186** | *Bacteroides ovatus* |
| 206 | *Bifidobacterium adolescentis/faecale* |
| **181,182,**195,196,201,202 | *Collinsella aerofaciens* |
| 200 | *Eubacterium rectale* |
| 187,188,189,190,191,192,193,194 | *Escherichia coli/fergusonii/vulneris/albertii/marmotae* |
| | *Shigella flexneri/sonnei/boydii/dysenteriae* |
| | *Brenneria alni* |
| 198 | *Hungatella effluvii* |

donor 3 and *Collinsella*, *Escherichia/Shigella*, *Eubacterium*, *Hungatella* and *Bifidobacterium* species in donor 4 (Figs. S1–S4, Table 1). The isolated species after enrichment correspond to the microbial community composition determined by Illumina sequencing, with the exception of *F. prausnitzii* and *Fusobacterium mortiferum*, which were enriched but could not be isolated on solid agar plates. *F. prausnitzii* is difficult to isolate due to its oxygen sensitivity (*Duncan et al., 2002*; *Khan, van Dijl & Harmsen, 2014*; *Lopez-Siles et al., 2017*). All isolation work was carried out in an anaerobic workstation and the YCFAG medium has been shown to support *F. prausnitzii* growth (*Duncan et al., 2002*; *Khan et al., 2012*). The solid agar media were, however, not pre-reduced in this experiment, as opposed to the liquid broth, which explains why *F. prausnitzii* could not be cultivated on YCFAG plates (*Holdeman et al., 1977*). The direct plating method resulted in the additional isolation of *Bacteroides*, *Bifidobacterium* and *Streptococcus* species (Table 1). This is in agreement with the results from the enrichment series of donor 2, showing a higher diversity, including *Bacteroides* species, after the first 24 h of incubation.

## Discussion of the experimental set-up

The enrichment and isolation of predominantly LAB, bifidobacteria and *Enterobacteriaceae* at first sight seems to contest the previous reports of wheat bran colonization by a subset of *Prevotella*, *Bacteroides* and *Clostridium* cluster XIVa organisms (*De Paepe et al., 2017*; *Duncan et al., 2016*; *Leitch et al., 2007*). There are, however, some important differences in experimental set-up that might account for the observed disparity and need to be addressed in future isolation procedures.

First of all, the comparable enrichment of LAB, bifidobacteria, *Streptococci* and *Faecalibacterium*, in the current study and in an isolation experiment using gluten as the major protein source, could indicate an inability of *Prevotella*, *Bacteroides* and *Clostridium* species to compete for wheat bran proteins. This would explain their limited growth compared to previous studies using a protein-rich medium containing peptone and yeast-extract (*De Paepe et al., 2017*; *De Paepe et al., 2018*).

Secondly, the 24 h incubation period in between transfers in the present study might have influenced the results. A detailed analysis of the time course of wheat bran colonization and fermentation (K De Paepe, J Verspreet, CM Courtin & T Van de Wiele, 2015, unpublished data) revealed a succession of bacterial taxa alternately dominating the community over a 72 h timespan. Early stages were dominated by *Enterobacteriaceae* and *Fusobacterium* species and characterized by a low butyrate production. After 48 h, the butyrate ratio increased, corresponding to donor-dependent proportional increases of *Bacteroides ovatus/stercoris*, *Prevotella copri* and *Firmicutes* species. We hypothesized that depletion of the easily digestible compounds induced a shift towards carbohydrate degrading specialists, possessing the enzymatic capacity to breakdown the complex molecules. This hypothesis is also valid with wheat bran as the sole nutrient source. Apart from residual starch, the more fermentable wheat bran components are located in the aleurone layer (*Amrein et al., 2003*; *Stevens & Selvendran, 1988*). Indeed, we observed a preferential microbial colonization and degradation of aleurone through SEM (Fig. 8). This aleurone degradation might be aided by the activity of wheat kernel associated endogenous or microbial enzymes, including xylanases. The latter remains speculative, as these enzymes might be deactivated upon gastro-intestinal passage and their activity can be inhibited by TAXI (*Triticum aestivum* xylanase inhibitor) and XIP (xylanase-inhibiting protein) xylanase inhibitors, which are also present in wheat bran.

Finally, the large extent of acidification due to the high wheat bran concentration in the first two donors (50 g L$^{-1}$) might be responsible for the unintentional selection for acidophilic or aciduric LAB.

In order to investigate the above hypotheses, it would be interesting to repeat the enrichment procedure with a medium containing an additional protein source, extending the incubation time in between passages to 48 h and ensuring a sufficient buffering capacity. This might result in the enrichment and isolation of a vastly different array of gut bacteria. In addition, for the direct plating strategy, the medium should be pre-reduced.

## CONCLUSIONS

Enrichment of the wheat bran-colonizing microbial community resulted in the isolation of a diverse set of *Lactobacillus, Bifidobacterium, Collinsella, Escherichia/Shigella, Pediococcus, Enterococcus, Klebsiella, Eubacterium and Hungatella* species. These isolated species were also found to be enriched on the wheat bran residue by next-generation amplicon sequencing, demonstrating that the proposed enrichment procedure is a sensible and efficient approach to isolate wheat bran-colonizing species. As insoluble wheat bran presented the sole carbohydrate and protein source, the isolated species should be capable of degrading and metabolizing some of the wheat bran constituents. Based on the metabolic capacity documented in literature, the isolates likely thrived on proteins and some residual easily fermentable starch in the current set-up, while in a protein-rich medium, we have previously revealed that the wheat bran-colonizing species possessed arabinoxylan and cellulose degrading enzymatic potential. Combining the new insights from this study with previous observations suggests that adhesion is a rather common trait among gut bacteria,

that the outcome of wheat bran colonization is determined by species competition and that external conditions such as pH and nutrient availability tip the balance in favor of the best-adapted species (*Macfarlane, Hopkins & Macfarlane, 2011*). It would, hence, be of interest to modify the experimental conditions, as outlined above, to enrich arabinoxylan and cellulose degrading species. Pure culture studies should then be performed to characterize the fermentation of wheat bran components by the obtained isolates and to further our understanding of the mechanisms of wheat bran attachment which could include Extracellular Polymeric Substances (EPS) or even a cellulosome enzyme system, in case of *R. champanellensis* and common adhesive features such as pili and fimbriae. Finally, the co-culturing of different isolates would be interesting to unravel cooperative and competitive interactions during substrate fermentation.

## ACKNOWLEDGEMENTS

We would like to acknowledge Veerle Rober for technical assistance and Jolien De Paepe for reviewing this manuscript.

### Funding

This research was funded by the Research Foundation Flanders (FWO, Grant id=IWT130028, title=SBO BRANDING) and the Special Research Fund (BOF) Concerted Research Actions (GOA, BOF17/GOA/032) from the Flemish Government. The Phenom SEM instrument was supported by funding from Research Foundation Flanders (FWO grant G031416N to FJRM). The Hercules Foundation provided financial support in the acquisition of the scanning electron microscope JEOL JSM-7100F equipped with the cryo-transfer system Quorum PP3010T (grant no. AUGE-09-029). There was no additional external funding received for this study. The funders had no role in study design, data collection and analysis, decision to publish, or preparation of the manuscript.

### Grant Disclosures

The following grant information was disclosed by the authors:
Research Foundation Flanders: IWT130028.
Special Research Fund.
Concerted Research Actions: BOF17/GOA/032.
Research Foundation Flanders: G031416N.
Hercules foundation: JEOL JSM-7100F.
Cryo-transfer system Quorum PP3010T: AUGE-09-029.

### Competing Interests

The authors declare there are no competing interests.
## Author Contributions

- Kim De Paepe conceived and designed the experiments, performed the experiments, analyzed the data, prepared figures and/or tables, authored or reviewed drafts of the paper, approved the final draft.
- Joran Verspreet, Mohammad Naser Rezaei and Silvia Hidalgo Martinez performed the experiments, contributed reagents/materials/analysis tools, approved the final draft.
- Filip Meysman and Koen Dewettinck contributed reagents/materials/analysis tools, authored or reviewed drafts of the paper, approved the final draft.
- Davy Van de Walle performed the experiments, contributed reagents/materials/analysis tools, authored or reviewed drafts of the paper, approved the final draft.
- Jeroen Raes contributed reagents/materials/analysis tools, approved the final draft.
- Christophe Courtin conceived and designed the experiments, contributed reagents/-materials/analysis tools, authored or reviewed drafts of the paper, approved the final draft.
- Tom Van de Wiele conceived and designed the experiments, authored or reviewed drafts of the paper, approved the final draft.

## Human Ethics

The following information was supplied relating to ethical approvals (i.e., approving body and any reference numbers):

Research incubation work with fecal microbiota from human origin was approved by the ethical committee of the Ghent University hospital under registration number B670201214538.

## Data Availability

The R code is available in Data S1 and S2 under the form of an RMarkdown file and the knitted PDF version. Raw SCFA and 16S rRNA gene amplicon sequencing data is included in the Data S3–S9. 16S rRNA gene Sanger sequences of the isolates are supplied as a compressed folder (Sanger_isolates.zip). Additionally, Data S10–S22 comprise (i) user defined functions, (ii) mothur reports with the closest 16S rRNA gene Sanger reference for each OTU obtained by 16S rRNA gene amplicon sequencing, (iii) OTU sequences obtained by 16S rRNA gene next-generation amplicon sequencing in FASTA format and (iv) RDP taxonomic annotation of the 16S rRNA gene Sanger sequences of the isolates, which are all imported in the RMarkdown file.

The sequence data is available in the NCBI database under accession number SRP091975.

## Supplemental Information

Supplemental information for this article can be found online at http://dx.doi.org/10.7717/peerj.6293#supplemental-information.

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
