# Peer review of "Isolation of wheat bran-colonizing and metabolizing species from the human fecal microbiota"

_PeerJ, doi:10.7717/peerj.6293_

## Round 0.1 · original submission · Minor Revisions

While the feedback of the reviewers was overall positive, they both commented on the possibility of contamination in the wheat bran and questioned the addition of glucose/the immediate transfer to rich medium. In addition, one reviewer questioned the high level of acetate instead of butyrate production. When addressing the reviewer comments, the authors should discuss these points in particular.

Reviewer 1 ·

Basic reporting

The paper by de Paepe et al. entitled “Isolation of wheat bran-colonising and metabolizing species from the human fecal microbiota” is well written and clearly set out. The methods section in particular is very detailed.

Experimental design

The use of non-autoclaved wheat bran is likely to have confounded the outcomes of the study. The study design and the approaches microbiology and molecular methods are well thought out.

Validity of the findings

The new bacterial isolates from wheat bran seem somewhat surprising given that wheat bran is considered to promote butyrate formation but the incubations mainly yielded acetate. This may be a feature of employing suboptimal conditions for the incubations.

Additional comments

Specific comments to the authors
Abstract. The abstract could be improved, for example line 43 Provide at least an example of “these” microorganisms. Lines 38-42 is essentially belong as discussion points rather than the final statement of the abstract.
Lines 46-49. Several of the references cited are not recent as dating back to 1980.
Line 55. Change to “.. pattern for..”.
Line 62-64. There are many different Bacteroides and Eubacterium species, and some of these may have a role in wheat bran metabolism whilst others are unlikely to do so, it would therefore be much better to state this at the species level.
Line 106-107. What was the reason for the reduced level of what bran?
Line 144. As the aim of the study was to isolate wheat bran degrading bacteria what was the rationale for using glucose as the added carbon source to the YCFA medium (given that the main sugars are arabinose and xylose)?
Line 168. Again what was the rationale for using glucose?
Line 195. Provide brief details of the DNA extraction method used here.
Line 300. I would suggest that using wheat bran that had not been autoclaved is likely to confound the study. Low level contaminants may have been missed. Would it have been feasible to irradiate the wheat as an alternative approach?
Line 313. Indicate the closest species, as certain bifid species may be expected to use wheat bran sugars whilst others would not.
Line 382. Change typo “Pediococus” to “Pediococcus”.

Reviewer 2 ·

Basic reporting

The manuscript entitled “Isolation of wheat bran-colonizing and metabolizing species from the human fecal microbiota” by De Paepe et al report the enrichment and isolation of different bacterial taxa from human faeces using the insoluble substrate wheat bran as sole nutrient source. The manuscript is clearly written, in professional and unambiguous English.

The manuscript is descriptive but provides interesting insights about the presence of wheat bran-metabolizing bacteria in the gastrointestinal tract. The methodology used can profit other scientists in the field. Indeed, the isolation of yet unculturable gut bacteria is an essential first step to increase our understanding of the microbiome-host interaction.

The size of the core manuscript seems rather extensive compared to the amount of results presented. To make it more concise, I would recommend the authors to move at least Table 1, 2, 3, 4 into supplementary tables, as well as Figures 7, 8, 9, 10, 11, 12 into supplementary figures.

Additional specific corrections:
Line 52 (comma missing): Macfarlane et al. (1997,2006)=> Macfarlane et al. (1997, 2006)
Line 55-56: colonization pattern insoluble substrates => colonization pattern of insoluble substrates
Line 103 (comma missing): The chemical composition is displayed in Fig. 1.The medium […] => The chemical composition is displayed in Fig. 1. The medium […]
Line 106 (comma missing): 0.1M phosphate buffer => 0.1 M phosphate buffer
Line 212: (LGCGenomics, (Teddington, Middlesex, UK) => (LGCGenomics, (Teddington, Middlesex, UK))
Line 201: primerpair => primer pair
Line 279: 30mA => 30 mA
Line 282: As an alternative to SEM microscopy, samples were also visualised using a Jeol JSM 7100F 283 scanning electron microscope (JEOL Ltd, Tokyo, Japan) => This sentence is rather confusing since authors proposed SEM as an alternative to SEM. I believe they meant the following: As an alternative to SEM microscopy, samples were also visualised via cryo-SEM using a Jeol JSM 7100F 283 scanning electron microscope (JEOL Ltd, Tokyo, Japan)
Line 346 (comma missing): in donors 1,3 and 4 => in donors 1, 3 and 4
Line 134 & 155: Please delete the brand of the anaerobic workstation => (GP-Campus, Jacomex, TCPS NV, Rotselaar, Belgium). It was already stated line 96.
Line 232 & 260: Please delete the version of R => (version 3.4.2 (2017-09-28) (R Core Team 2016)). It was already stated line 181.
Line 470: aleurone through Scanning Electron Microscopy (SEM) (Fig. 8) => aleurone through SEM (Fig. 8)

Experimental design

Authors have made a good job at providing detailed information so that other investigators could reproduce these experiments. Few experiments remains however unclear:

Comments regarding the wheat-bran preparation:
Since wheat bran is the core nutrient for enriching and isolating bacteria, authors should be more explicit regarding the preparation (=pre-digestion) and chemical analysis of wheat-bran (line 101). It is currently slightly confusing:

Authors refer to the paper of De Paepe et al. (2017) which is a modification of the paper from Minekus et al., (2014).
In De Paepe et al (2017) the legend of Table 1 states: “The methods used for wheat bran micronization and to determine the wheat bran composition are described in the Supporting Information”. However, no supporting information is available in that regards. When reading the experimental procedure from De Paepe et al (2017), it is also unclear how and when the analysis of the chemical composition of wheat bran has been performed. It could be pre-digestion with this statement: “Commercial wheat bran with chemical composition displayed in Table 1”. Or post-digestion with this statement “Insoluble undigested material was centrifuged and lyophilized to preserve until chemical analysis or batch incubation in hungate tubes”.

Therefore, in regards to the current manuscript, the chemicals used and the different steps/reactions of the digestion steps could be more clearly stated (for instance within the section supplementary material). The method use for determining the chemical composition as well. In Figure 1, please clarify the “state of wheat bran” used. If available, it would be valuable to provide the readers with the chemical analysis of both before and after-digestion.

Line 367: Authors stated that the bacteria retrieved from the enrichment were unexpected due to their lack of enzymes for the degradation of cellulose and arabinoxylan polymers. Various enzymes + mucin were added during the digestion of wheat-bran, but their fate is unclear at the end of digestion. Could these biomolecules be responsible for the unexpected growth?


Comments on SEM and cryo-SEM:
Line 470: Authors state a preferential microbial colonization and degradation of aleurone based on SEM pictures. This statement is hard to verify with the pictures provided. The scale of Fig. 8A is greatly different than those of Fig. 8B-F. Bacteria might thus be hard to visualize.
Are there any reasons for authors to use Cryo-SEM (Fig. 7A) vs SEM (Fig. 7B) to compare the effect of digestion? Two different scales were also used.

Additional specific comments:
Line 394-396: Do authors refer to another article or to personal data for the statement regarding the presence of enzymatic activity after pre-digestion? Please clarify.
Line 400-401: Do authors refer to another article or to personal data regarding the capacity of F. prausnitzii to degrade wheat bran in pure culture? Please clarify

Validity of the findings

Comment on the isolation of wheat-bran metabolizing bacteria:
Authors have first used “wheat-bran agar media” to isolate colonies from fecal slurry. The colonies were then directly transferred onto a nutrient-rich agar medium (YCFAG). Are there any reasons for authors not to transfer colonies on the same “wheat-bran agar media" for a second or even a third time to ensure that growth was due to the utilization of wheat-bran, and not because of remaining nutrients from the fecal slurry?

Colonies from YCFAG agar were subsequently transferred in YCFAG broth. Have authors tried to grow the pure cultures from Table 5 in the “wheat-bran broth” to confirm their capacity to use wheat-bran as sole nutrient source?

Additional comments

A suggestion to authors: They stated (line 299) that wheat bran was not autoclaved to avoid structural modifications. This is biologically sounded, but it seems that bacteria have grown in the control. In the future, authors could try to UV the wheat bran for couple of minutes (with shaking to expose all surfaces) in an attempt to remove, or at least reduce, the risk of contamination. Short exposure to UV should not impact its composition.

---

## Round 0.2 · accepted · Accept

All reviewer remarks were addressed in an adequate manner. I therefore recommend the publication of this work.

#